# In Vitro Evaluation of the Regenerative Potential of Autologous Platelet-Rich Fibrin (PRF) on Human Primary Periodontal Ligament Cells

**DOI:** 10.3390/ijms26199459

**Published:** 2025-09-27

**Authors:** Eva Dohle, Marlene Quernheim, Robert Sader, Shahram Ghanaati

**Affiliations:** 1Frankfurt Orofacial Regenerative Medicine (FORM), Department for Oral, Cranio-Maxillofacial and Facial Plastic Surgery, Medical Center, Johann Wolfgang Goethe University, 60596 Frankfurt, Germany; 2Academy for Biological Innovations in Surgery (ABIS e.V.) Formerly Known as Society for Blood Concentrates and Biomaterials e.V. (SBCB e.V.), 60435 Frankfurt, Germany

**Keywords:** periodontal ligament cells, platelet-rich fibrin, regeneration, osteogenic differentiation

## Abstract

Periodontitis is a prevalent condition that leads to the destruction of periodontal tissue, making the regeneration of the periodontium a key focus in dental research. In this context, periodontal ligament cells (PDLCs) are particularly interesting due to their stem cell-like properties, including the ability to differentiate into various cell types and further contribute to tissue repair. This study aimed to isolate and characterize primary human PDLCs and examine the effects of both indirect and direct treatment with the blood concentrate platelet-rich fibrin (PRF), with particular focus on how PRF influences cell proliferation and differentiation. PDLCs were treated with PRF prepared using a low relative centrifugal force (600 rpm) either directly through a conditioned medium or indirectly using trans-well filter systems. The impact of PRF on PDLC proliferation and differentiation was assessed through viability assays, alkaline phosphatase assays, gene and protein expression analyses, and immunofluorescence. PDLCs exhibited cellular markers characteristic of stem cell-like cells. In addition, PRF treatment was found to suppress cell proliferation while concurrently promoting osteogenic differentiation and increase factors important for tissue regeneration. These effects were more pronounced when the cells were directly treated with PRF-conditioned medium compared to indirect treatment. Our findings support the hypothesis that PRF serves as a biologically active reservoir of growth factors that modulate PDLC behavior and might create a microenvironment favorable for periodontal repair.

## 1. Introduction

Periodontitis is a widespread disease and, due to the destruction of the periodontium, is the main cause of tooth loss in adult [1,2]. The primary therapeutic objective in the management of periodontitis is the elimination of inflammation and the prevention of further tissue degradation [2]. In this context, regenerative approaches aimed at restoring lost periodontal structures to achieve periodontal regeneration, while the importance of stem cell-like periodontal ligament cells (PDLCs) has steadily increased [3,4,5]. Among the various cellular components involved in periodontal regeneration, periodontal ligament cells (PDLCs) play a central role. These cells constitute a heterogeneous population, encompassing fibroblasts, progenitor cells, and mesenchymal stem cell-like subpopulations that are essential for periodontal repair and regeneration [5,6]. Notably, periodontal stem cells exhibit remarkable differentiation potential, capable of transforming into fibroblasts, cementoblasts, osteoblasts, chondrocytes, and adipocytes, indicating a multipotent nature [7,8]. With high proliferation and differentiation capacity, hPDLCs play a vital role in maintaining periodontal tissue homeostasis, balance, and regeneration. These attributes make periodontal stem and progenitor cells key targets in various therapeutic strategies aimed at promoting their proliferation and migration into wound sites. Despite substantial progress in characterizing hPDLCs, a comprehensive understanding of the signaling mechanisms and environmental cues governing their regenerative potential remains incomplete [5,9]. Increasing evidence suggests that successful periodontal healing is orchestrated through complex interactions between resident and recruited immune cells, extracellular matrix (ECM) components, and locally released growth factors and cytokines.

In recent years, autologous blood-derived biomaterials have emerged as promising tools to enhance tissue regeneration by providing a concentrated source of growth factors and wound-healing mediators at the site of injury. Among these, Platelet-Rich Fibrin (PRF) has gained substantial interest in both dental and broader regenerative medicine fields [10,11,12]. PRF is a second-generation, purely autologous platelet concentrate that requires no external additives [10,13]. The absence of exogenous agents allows for the natural formation of a three-dimensional fibrin scaffold that entraps a high concentration of platelets, leukocytes, and a wide array of bioactive molecules. Key growth factors present in PRF include Platelet-Derived Growth Factor (PDGF), Transforming Growth Factor Beta (TGF-β), Epidermal Growth Factor (EGF), and Vascular Endothelial Growth Factor (VEGF), all of which play critical roles in modulating angiogenesis, cell proliferation, differentiation, and extracellular matrix remodeling during tissue regeneration [14,15,16]. The refinement of PRF preparation protocols, particularly the development of the low-speed centrifugation concept (LSCC), has further enhanced the biological efficacy of PRF by reducing the relative centrifugal force (RCF), thereby increasing the retention of cells and growth factors within the matrix [17]. Although previous studies have demonstrated that PRF exerts beneficial effects on PDLCs, including enhanced proliferation and differentiation, the capacity of PRF to achieve complete periodontal regeneration remains under active investigation [12,18,19].

Given the existing gaps in knowledge, the present study aims to evaluate the biological effects of PRF on hPDLCs in vitro, with a specific focus on comparing the outcomes of two distinct PRF application protocols. In this study, PRF was either applied directly using trans-well inserts or indirectly through the use of PRF-conditioned media. The influence of these treatments on hPDLCs was assessed at two time points, 3 and 7 days, using a combination of assays evaluating cell viability, alkaline phosphatase (ALP) activity, osteogenic differentiation, and the expression of key genes and proteins associated with stemness and the cell cycle. Through this comparative analysis, the study seeks to elucidate both the general impact of PRF on hPDLC behavior and the potential differential effects mediated by varying modes of PRF application, thereby contributing to the optimization of PRF-based regenerative strategies for periodontal therapy.

## 2. Results

### 2.1. Characterization of Periodontal Ligament Cells

Immunofluorescence staining was employed to assess the expression of key mesenchymal and periodontal ligament-associated markers in human primary periodontal ligament cells (hPDLCs). The results confirmed that hPDLCs were positive for vimentin, CD105, CD73, and CD90, indicating their mesenchymal origin and stem cell-like phenotype (Figure 1A–D). Morphologically, hPDLCs exhibited a heterogeneous population of cells with polygonal, spindle-shaped, and star-shaped appearances, often displaying prominent, elongated cytoplasmic extensions, which is characteristic of fibroblast-like and progenitor cell morphology. To further investigate the molecular profile of hPDLCs, quantitative gene expression analyses were performed for a range of markers associated with osteogenesis, cell proliferation, cell cycle regulation, stemness, and cell adhesion. These profiles were compared to those of human primary fibroblasts (hFBs) to determine the distinct molecular characteristics of hPDLCs (Figure 2). The osteogenic marker alkaline phosphatase (ALP) was significantly downregulated in hPDLCs compared to hFBs (Figure 2A), suggesting a reduced baseline osteogenic activity under non-differentiating conditions. In contrast, COL1A1, a gene encoding type I collagen, a key component of the extracellular matrix in periodontal tissues, was significantly upregulated in hPDLCs (Figure 2B). In terms of cell proliferation and cell cycle regulation, hPDLCs demonstrated significantly higher expression of cyclin D1 (CCND1) (Figure 2C) and the Ki67 proliferation marker (Figure 2D) compared to hFBs. Additionally, expression of telomerase reverse transcriptase (TERT), which is associated with extended proliferative potential and cellular longevity, was also significantly upregulated in hPDLCs (Figure 2E). Analysis of insulin-like growth factor 2 (IGF-2), a factor known for its mitogenic and anti-apoptotic effects, revealed a significant upregulation in hPDLCs relative to hFBs (Figure 2F). Interestingly, CXCR4, a chemokine receptor involved in stem cell homing, migration, and tissue regeneration, was found to be significantly downregulated in hPDLCs compared to hFBs (Figure 2G). In contrast, SOX2, a transcription factor crucial for maintaining pluripotency and the undifferentiated state of stem cells, was significantly upregulated in hPDLCs (Figure 2H). Expression of CD146 (MCAM), a marker commonly associated with mesenchymal stem cells (MSCs) and correlated with enhanced differentiation potential, was significantly lower in hPDLCs compared to hFBs (Figure 2I). Finally, the gene expression levels of key integrins, including integrin α2, integrin β1, and integrin αV, were significantly upregulated in hPDLCs compared to hFBs (Figure 2J–K). These integrins are known to play essential roles in cell–matrix interactions, signal transduction, and the regulation of cellular functions such as proliferation, differentiation, and apoptosis.

### 2.2. Effect of Different PRF Treatment Methods on hPDLCs

In the following, hPDLCs that were indirectly treated with PRF in trans-wells are referred to as group ‘A’ (hPDLCs + PRF) and hPDLCs that were treated with a PRF-conditioned medium are referred to as group ‘B’ (hPDLCs + PRF cond.). The medium of group A alone is therefore abbreviated as ‘PRF’ and the medium of group B as ‘PRF cond.’. Differences and similarities between the various treatments were described both between the treatment and control groups and between the two different treatment methods (A and B).

#### 2.2.1. PRF-Mediated Effect on hPDLCs Cell Viability

To evaluate the impact of PRF on cellular viability and metabolic activity, the MTS assay was performed on days 3 and 7 post-treatment (Figure 3). The assay results were expressed as relative cell viability compared to the untreated control group (hPDLCs cultured in standard growth medium). After 3 days of treatment, Group A (hPDLCs + PRF) exhibited a significant reduction in cell viability compared to the control group (Figure 3A), suggesting that indirect PRF exposure via trans-wells may initially exert a suppressive effect on cell metabolism or proliferation. In contrast, Group B (hPDLCs + PRF cond.) displayed a slight, though not statistically significant, increase in cell viability relative to the control (Figure 3C). By day 7, cell viability in Group A had increased compared to day 3 level but remained significantly lower than the untreated control (Figure 3B). Interestingly, Group B, which initially showed slightly increased viability at day 3, demonstrated a significant decrease in viability after 7 days of PRF-conditioned medium treatment compared to the control (Figure 3D).

#### 2.2.2. Evaluation of Protein Concentration in Cell Culture Supernatants

To compare the protein expression of PRF treated and untreated hPDLCs, the concentrations of key soluble factors involved in osteogenesis, angiogenesis, inflammation, and tissue remodeling were quantified over time. The concentrations of osteoprotegerin (OPG), vascular endothelial growth factor (VEGF), interleukin-6 (IL-6), and transforming growth factor beta (TGF-β) were measured in culture supernatants collected on days 3 and 7, and results were expressed as absolute concentrations (Figure 4A–D). Between day 3 and day 7, all study groups including the untreated control, Group A (hPDLCs + PRF), and Group B (hPDLCs + PRF-conditioned medium) demonstrated a comparable increase in absolute OPG concentrations (Figure 4A) indicating a baseline osteogenic signaling activity that is not markedly altered by either form of PRF treatment within the observed timeframe. In the cumulative profile of VEGF secretion, all groups exhibited an increased VEGF concentration between days 3 and 7 (Figure 4B). Notably, the VEGF concentration was higher when hPDLCs were treated with PRF indirectly (group A) compared to the treatment with the conditioned medium (group B) and the control. IL-6 secretion showed group-specific differences. While all groups demonstrated an increase over time, Group B (PRF-conditioned medium) exhibited the most pronounced rise in absolute IL-6 levels between day 3 and day 7 (Figure 4C), followed by Group A and the control group. The cumulative concentration of TGF-β increased in all groups between the two time points. However, Group B showed a significantly greater increase in absolute TGF-β levels compared to Group A and the untreated control (Figure 4D) suggesting that direct exposure to PRF-conditioned medium might provide a more potent stimulus for TGF-β secretion by hPDLCs than indirect PRF treatment.

#### 2.2.3. PRF-Mediated Effect on Gene Expression of Differentiation, Proliferation and Stem Cell Associated Markers in hPDLCs

To assess the impact of PRF treatment on the gene expression of markers associated with osteogenic differentiation, cell proliferation, stemness, and signal transduction, quantitative real-time PCR (qRT-PCR) was conducted after 3 and 7 days of treatment. The following genes were analyzed: ALP, COL1A1, IGF-2, Ki67, CCND1 and CD146. Expression levels were normalized to GAPDH, and results were expressed as relative expression (RQ) compared to untreated hPDLCs (RQ = 1) (Figure 5). The results indicate that PRF treatment modulates gene expression in hPDLCs in a time- and delivery method-dependent manner. In general, PRF-conditioned medium (Group B) induced a stronger osteogenic and stemness-related transcriptional response while also more significantly suppressing proliferation compared to indirect PRF exposure (Group A). ALP expression was elevated in both treatment groups after 3 days but did not reach statistical significance (Figure 5A). After 7 days of treatment, ALP expression levels in Group A decreased and approached control levels, while Group B exhibited a significantly higher ALP expression, indicating enhanced osteogenic activity. Conversely, COL1A1 expression was consistently downregulated in both treatment groups at all time points, with Group B showing the greatest reduction, particularly after 7 days (Figure 5B). IGF-2 expression was significantly upregulated in Group A after 3 days, with no significant change in Group B (Figure 5C). However, by day 7, Group B exhibited a marked upregulation (~6-fold) of IGF-2, indicating a delayed but robust activation of IGF signaling. Ki67, a proliferation marker, was significantly downregulated when applying PRF in both groups throughout the study (Figure 5D). While both groups showed similar suppression at day 3, Group A maintained slightly higher Ki67 expression than Group B by day 7. Furthermore, CCND1 expression was slightly downregulated in Group A and significantly decreased in Group B at both time points (Figure 5E). CD146 expression, a marker of mesenchymal stem cells, was downregulated in both groups at both time points, with statistical significance for Group A at day 3 and Group B at day 7 (Figure 5F).

#### 2.2.4. Alkaline Phosphatase Activity in PRF Treated hPDLCs

To investigate the effects of PRF treatment on bone metabolism, ALP activity was quantified in supernatants of treated and untreated hPDLCs after 3 and 7. In addition, the respective ALP activity was determined in the sole medium of the different treatment methods (A and B), and results were presented as relative ALP activity in relation to the untreated control group (Figure 6). The relative ALP activity in supernatants of group B was higher and statistically significant after 3 and 7 days of PRF treatment compared to the control (C). The results of group A showed a similar ALP activity level as supernatants of untreated hPDLCs (A,B). The pure medium of group B showed almost double the relative ALP activity of the medium of group A and corresponded to the activity level of the untreated cells (A–D).

## 3. Discussion

In recent years, increasing emphasis has been placed on regenerative strategies aimed at restoring the lost periodontal structures, thereby achieving true periodontal regeneration. Among the cellular components involved in this complex regenerative process, periodontal ligament cells (PDLCs) have emerged as key players. PDLCs represent a heterogeneous cell population, including fibroblasts, progenitor cells, and mesenchymal stem cell-like subpopulations, all of which contribute critically to the repair and regeneration of periodontal tissues. To enhance wound healing and regeneration, blood concentrates such as platelet-rich fibrin (PRF) have attracted significant attention in dentistry and regenerative medicine due to their provision of essential immune cells and growth factors with regenerative potential. This study aimed to further characterize human primary periodontal ligament cells (hPDLCs) and investigate the effects of PRF on hPDLCs, with a particular focus on its influence on proliferation and differentiation processes. By examining the impact of two different PRF treatments on hPDLCs, our findings demonstrated that PRF can support and modulate various processes, including osteogenic differentiation and immunomodulation.

The characterization of hPDLCs confirmed their heterogeneous nature, including the presence of mesenchymal stem cell markers such as CD73, CD90, and CD105, alongside fibroblast-like morphology. The similarity of hPDLCs to mesenchymal stem cells are in agreement with the results of many other studies on the characterization of PDLCs [5,9,20]. Their fibroblast-like morphology has also been described in various studies and is consistent with our results [21,22]. The presence of vimentin, a cytoskeletal protein typically expressed in mesenchymal cells, supports the mesenchymal origin of the isolated hPDLCs [23]. CD105, a surface glycoprotein expressed on endothelial cells, PDLCs, and mesenchymal stem cells (MSCs), further highlights the heterogeneous nature of the cell population [24,25,26]. The membrane protein CD90 is one of the best-known stem cell markers and could be clearly visualized on hPDLCs in immunofluorescence staining [27]. In mesenchymal stem cells, a connection between CD73 expression and their different regenerative properties is supposed. hPDLCs were CD73 positive in immunofluorescence, which could speak in favor of their pronounced regenerative potential. Compared to human fibroblasts, hPDLCs demonstrated strong expression of markers associated with osteogenesis, cell proliferation, cell cycle regulation, stemness genes, including COL-1, CCND1, Ki67, TERT, and IGF-2, highlighting their high division rate and regenerative potential. hPDLCs showed lower ALP expression but significantly higher collagen production, suggesting their active role in tissue repair. The significantly lower ALP gene expression in hPDLCs than in hFBs suggests an active involvement of hFBs in repair processes [28]. Interestingly, the higher gene expression of COL-1 in hPDLCs compared to hFBs was consistent with another study where PDLCs were also shown to produce more collagen than gingival fibroblasts, suggesting an impressive regenerative capacity of hPDLCs [29]. The significantly higher relative expression of the genes CCND1, Ki67, TERT and IGF-2 in hPDLCs speaks in favor of their high division rate and their stem cell-like properties, which have also been described in other studies [6,30,31]. The SOX-2 gene expression confirms the presumed stem cell character of hPDLCs and are consistent with the results of other studies [32]. However, the lower relative expression of the CXCR4 and CD146 genes in hPDLCs than in hFBs could also indicate their limited ability to acquire further stem cells in the unstimulated state. The relative expression of the integrin genes ITG α-2, ITG ß-1 and ITG α-V in hPDLCs was significantly higher than the expression level of hFBs. Strong expression of these genes in hPDLCs was also found in other studies and indicates increased cell adhesion, migration and interaction with the environment, which are prerequisites for wound healing and tissue regeneration [33].

PRF treatment influenced PDLCs in a method-dependent manner. Direct exposure to PRF resulted in reduced proliferation, accompanied by enhanced ALP expression and activity, IGF-2 upregulation, and increased VEGF and TGF-β secretion. Collectively, these changes point toward an osteogenic shift, consistent with the concept that PRF promotes differentiation at the expense of proliferation. This transition is particularly relevant for tissue regeneration, where differentiation into osteoblast-like cells and stimulation of angiogenic and immunomodulatory pathways are more beneficial than continued cell division. The effect of PRF on the viability of hPDLCs was analyzed on the 3rd and 7th day after treatment indicating a suppressive effect of PRF on the metabolic activity and proliferation of hPDLCs. This is in contrast to other studies since the majority of literature reports increased cell proliferation after PRF treatment, for which the strongly mitogenic PDGF is considered to be responsible [34,35,36]. Nevertheless, there are also studies with contrary results that support our findings [37,38]. Accordingly, the effect on cell proliferation appears to be dependent on the concentration of PRF used [39]. It can be assumed that the concentration of the effective components of PRF in treatment groups A and B of the present study differed due to the methodology and influenced the result. In group A, a delayed proliferative effect may have occurred due to the indirect treatment with PRF, while the reduction in cell viability over time in group B indicates differentiation processes in hPDLCs. A decrease in cell proliferation would support the hypothesis of potential PRF-induced differentiation. Our results initially showed no clear effect of PRF treatment on OPG protein expression in hPDLCs since no higher concentration was detectable in the supernatants of the treated cells than in those of the untreated control. The RANK/RANKL/OPG system functions as a dynamic regulatory unit and changes in free OPG concentration alone may not fully reflect its biological activity. Increased binding of OPG to RANKL could have led to a reduction in the OPG concentration in the medium and signaled osteogenic differentiation to the cells [40]. RANKL and OPG are both produced by osteoblasts and bone resorption is activated by RANKL binding to RANK on osteoclast precursors during osteoclastogenesis. The physiological inhibitor of RANKL, OPG, blocks RANKL from binding to RANK, inhibiting osteoclastogenesis. Various studies support the hypothesis that PRF, by positively influencing OPG expression, can maintain the balance of bone-degrading and osteogenic differentiation and can achieve promising results as an adjuvant therapeutic agent [41]. A shift in the balance between RANKL and OPG, rather than an absolute increase in OPG, could be the key determinant of the observed differentiation effects [42]. Future studies should therefore not only measure OPG concentration but also assess RANKL and the RANKL/OPG ratio, and downstream signaling events to provide a more comprehensive understanding of the influence of PRF on this pathway in the context of the present study. Measurement of VEGF in cell culture supernatants showed a stimulating effect of PRF, which was more pronounced in group B. Other in vitro and in vivo studies also showed an induction of VEGF expression after PRF treatment [43]. Due to the promoting effects of vascular endothelial growth factor on angiogenesis and osteogenesis, a high VEGF concentration in the healing phase can contribute to bone regeneration. A marked increase in IL-6 concentration after PRF treatment can be attributed to production of IL-6 in white blood cells [44]. Both pro- and anti-inflammatory signaling pathways of IL-6 are essential for wound healing: in the initial inflammatory phase, the pro-inflammatory effects facilitate wound cleansing by recruiting macrophages and other immune cells, whereas in the later healing stages, its anti-inflammatory effects prevent excessive immune reactions and promote regenerative processes [45]. Similarly, the high TGF-ß concentrations observed in PRF-treated cultures align with the well-documented abundance of this growth factor in PRF [46]. In group B, the high TGF-ß concentration in the supernatant might suggest active receptor binding and utilization by hPDLCs indicating a TGF-ß-mediated role in proliferation, differentiation, and immunomodulation of the cells [7].

With regard to osteogenic differentiation, PRF treatment resulted in an upregulation of relative ALP gene expression and enhanced ALP activity, suggesting an early differentiation of hPDLCs into osteoblast-like cells. These results are consistent with previous studies that reported similar effects [47]. PRF-induced ALP activity also enhances the supply of inorganic phosphate, which contributes to bone regeneration in vivo [48]. Interestingly, COL-1 gene expression was consistently downregulated after PRF treatment compared to controls at all time points. Since excessive collagen expression can lead to fibrosis and scar formation, the observed downregulation suggests that PRF may exert an anti-fibrotic effect, thereby favoring the regeneration of structurally balanced tissue [29,49]. In addition, IGF-2 expression was significantly induced by PRF, particularly in group A at day 3 and in group B at day 7. IGF-2, a fetal growth factor involved in cell growth and differentiation, also contributes to bone formation and repair in adults [50]. Although this pathway is not yet fully understood, the data suggest that PRF promotes bone formation in hPDLCs via IGF-2 upregulation. The expression analysis of proliferation and cell cycle markers revealed that PRF suppressed CCND1 and Ki67 expression, suggesting reduced cell division. Although this finding contrasts with some studies that reported proliferative effects of PRF, it can be explained by a shift in cellular behavior from proliferation toward differentiation. This hypothesis is supported by the concurrent upregulation of osteogenic markers and is consistent with observations from other groups [51,52]. PRF treatment also influenced the expression of stem cell markers. Direct PRF application upregulated CXCR4, a gene known to drive differentiation into osteoblasts and other lineages [53]. CD146 expression was consistently downregulated after PRF treatment, which may indicate that hPDLCs were transitioning from a stem-like state to a more specialized phenotype [54]. Together, our results support the hypothesis that PRF promotes osteogenic differentiation by regulating stemness and differentiation-related gene expression Direct PRF treatment (group B) consistently produced more pronounced effects on ALP activity, gene expression, and protein levels than indirect treatment (group A). This may be due to higher and more uniform growth factor release, as well as the possible presence of leukocytes in group B, which could contribute to cytokine release. By contrast, the trans-well system used in group A limited direct interaction and may have reduced the bioactivity of PRF. In summary, the findings demonstrate that PRF is a promising autologous biomaterial for regenerative applications. By combining pro-osteogenic, immunomodulatory, and pro-angiogenic properties, PRF supports wound healing and tissue regeneration. Importantly, direct PRF treatment produced stronger and more consistent effects than indirect treatment, and therefore should be prioritized in future experimental and clinical studies. Nevertheless, this study has limitations. The in vitro design cannot fully replicate the complexity of in vivo tissue environments, and variability among donor-derived PDLCs may affect the generalizability of the results. Further research should focus on clarifying the molecular pathways underlying PRF-mediated osteogenic effects and on exploring synergistic strategies that combine PRF with biomimetic scaffolds or additional growth factors. Preclinical and clinical studies will be essential to confirm these findings, standardize PRF preparation protocols, and establish its role as a predictable adjunct in periodontal regenerative therapy.

## 4. Materials and Methods

### 4.1. Ethical Statement

All cells that were used for this study were obtained from excess tissue and their application was in accordance with the principle of informed consent and approved by the responsible Ethics Commission of the state Hessen, Germany. In addition, the application of PRF in this study was approved by the responsible Ethics Commission of the state of Hessen, Germany (265/17) and all donors gave informed consent to the use of their blood for study purposes.

### 4.2. Primary Cells

Human primary periodontal ligament cells (hPDLCs) were isolated from the root surfaces of surgically removed wisdom teeth. Tissue samples were stored in cell culture medium (DMEM with 10% fetal calf serum (FCS Sigma-Aldrich, St. Louis, MO, USA) and 1% penicillin/streptomycin (P/S, Sigma-Aldrich, St. Louis, MO, USA)) and directly washed with PBS. Using a sterile scalpel and tweezers, tissue from the middle third of the root surfaces was dissected into a Petri dish with PBS. The tissue suspension was centrifuged at 1200 rpm for 5 min, the supernatant removed, and the pellet resuspended in 2 mL of 0.04% trypsin solution for enzymatic digestion. After 2 h of incubation at 37 °C, the reaction was stopped with 2 mL of trypsin neutralization solution (TNS, PromoCell, Heidelberg, Germany). The suspension was filtered through a 100 µm cell strainer, centrifuged again, and the pellet was resuspended in 2 mL of DMEM (10% FCS, 1% P/S). Cells were used up to passage four. Human fibroblasts (hFBs) used in this study were isolated from human excess tissue. hFBs were isolated with human dermal microvascular endothelial cells from excess tissue of cleft lip and palate reconstructions of adolescent patients and subsequently separated. Following an established protocol, the cells were enzymatically digested in several steps and isolated by magnetic cell separation for the endothelial cell-specific CD31. As CD31-negative cell fraction, primary hFBs were subsequently cultured in DMEM (10% FCS, 1% P/S) at 37 °C in a humidified atmosphere. Cells were used until the third passage.

### 4.3. Preparation of PRF

Whole blood was collected from at least three donors of all genders who provided informed consent to participate in the study. The donors were healthy, free from infectious diseases, not taking anticoagulants, and did not consume alcohol or nicotine. Their ages ranged from 20 to 50 years. For PRF preparation, 10 mL of peripheral blood was drawn from the median cubital vein and collected into glass- or plastic-coated PRF tubes (Pro Cell, Nice, France). The blood was immediately centrifuged in a Duo centrifuge (Mectron, Cologne, France) with a fixed-angle rotor (110 mm radius). Following the LSCC (low-speed centrifugation concept) protocol, centrifugation was performed at 600 rpm for 8 min [17]. After centrifugation, the clear, yellow-colored liquid PRF layer was carefully collected using a syringe, while the solid PRF clot was retrieved with tweezers. Liquid PRF from each donor was transferred into sterile 15 mL tubes for homogenization and directly used in cell culture experiments. Solid PRF clots were cultured in DMEM (10% FCS, 1% P/S) at 37 °C in a humidified atmosphere, and the PRF conditioned medium was collected on the fourth day, individually for each donor.

### 4.4. Indirect and Direct PRF Treatment of hPDLCs

For indirect (group A) as well as for direct (group B) PRF application 50.000 hPDLCs per well were pre-seeded in 24-well plates and cultured in DMEM containing 10% FCS and 1% penicillin/streptomycin for 24 h. For the separation of PRF and the pre-seeded cells, trans-well inserts with a pore size of 0.4 μm (Greiner Bio-One, Kremsmünster, Germany) were utilized for indirect PRF treatment allowing only growth factors and signaling molecules to pass through the membrane, but not the cells. Therefore, the trans-wells were placed in the pre-seeded 24-well plates and 200 μL of liquid PRF was added directly into the trans-wells. Wells containing solely hPDLCs without the addition of PRF served as controls. After the clotting process of PRF was completed, 1 mL of DMEM (10% FCS, 1% P/S) was added to the lower compartment of the wells and changed after 3 days. An additional 200 μL of medium was added to the inner part of each trans-well to prevent the PRF from drying. For direct PRF application, supernatants of cultured PRF clots were used for cultivation of pre-seeded hPDLCs. The conditioned medium was changed and renewed after 3 days. Cells and cell/PRF complexes were cultured for 7 days. Supernatants were collected after 3 and 7 days and stored for ELISA and ALP-activity assay. Finally, the cells were analyzed for cell viability after 3 and 7 days as well as fixed for immunofluorescence staining and processed for gene expression analyses after 3 and 7 days of PRF treatment.

### 4.5. Cell Viability Assay

The effect of indirect and direct PRF treatment on cell viability of hPDLCs was examined after 3 and 7 days of cultivation. After washing the wells with PBS, DMEM (10% FCS, 1% P/S) and 100 µL CellTiter 96ⓇAQueous One Solution Reagent (Promega, Madison, WI, USA) were added followed by an incubation period of 2 h at 37 °C. A total of 100 μL from each well was transferred to a separate 96-well plate and absorbance was measured at 490 nm in a microplate reader (Infinite M299, Tecan, Zurich, Switzerland).

### 4.6. Immunofluorescence Staining

For immunofluorescence staining cells were fixed in 4% buffered formalin (Roti-Histofix 4%, acid-free pH7, Carl-Roth, Germany) washed 3 times with phosphate-buffered saline (PBS), permeabilized for intracellular antigens with 0.1% Triton-X100/PBS and incubated with the specific primary antibody (CD105, Vimentin, CD73, CD90) diluted in 1% BSA/PBS solution (1:30 for CD105 (Agilent Dako, Santa Clara, CA, USA), 1:100 for Vimentin (Agilent Dako, Santa Clara, CA, USA), CD73 (Abcam, Cambridge, UK), CD90 (Biozol, Eching, Germany)), for 1 h at RT (room temperature). Fluorescently labeled secondary antibodies (Alexa fluor 488 anti-mouse, Thermo Fisher Scientific, Karlsruhe, Germany) were diluted in 1% BSA/PBS and incubated in a dark environment for 1 h at RT. Thereafter, the cell nuclei were counterstained with DAPI diluted in 1% BSA/PBS (1:1000). The fluorescent D-LEDI LED illumination system of the Eclipse Ni/E with the Nikon DS-Ri2 camera (Nikon eclipse Ni/E, Düsseldorf, Germany) was used for evaluation.

### 4.7. Enzyme-Linked Immunosorbent Assay (ELISA)

The collected cell culture supernatants were quantified for relative protein concentrations of VEGF, OPG, IL-6 and TGF-β using ELISA-DuoSet development system (R&D Systems) according to the manufacturer’s instructions. A microplate reader (Infinite M200, Tecan, Zurich) detected the optical density of each well at a wavelength of 450 nm.

### 4.8. RNA Isolation and Gene Expression Analyses

RNA isolation was performed using the RNeasy Micro Kit (Qiagen, Hilden, Germany) according to the manufacturer’s instructions. Using the standard protocol of Qiagen’s Omniscript reverse transcription kit, 1 μg of the extracted RNA per sample was transcribed into complementary DNA (cDNA). The relative gene expression of osteogenic differentiation factors, cell proliferation or stem cell markers and cell adhesion molecules was analyzed using specific primers for ALP, Col-1, CCND1, Ki67, TERT, IGF-2, CXCR4, SOX-2, CD146, ITG-α1, ITG-ß1 and ITG-αV. For quantitative reverse transcription polymerase chain reaction (qRT-PCR), the RT-PCR cycler StepOnePlus by Applied Biosystems was implemented. SYBR^®^ green was utilized as a DNA-binding fluorescent dye. Analyses were performed in triplicates with a standardized cycler program: 94 °C for 2 min, 94 °C for 15 s, and 60 °C for 1 min. The reactions were run for 40 cycles. Glyceraldehyde-3-phosphate dehydrogenase (GAPDH) served as the endogenous standard. The relative gene expression was determined using the ∆∆Ct method. As a reference value, the control cultures were set to one.

### 4.9. ALP-Activity Assay

The alkaline phosphatase (ALP) assay (Abcam, Cambridge, UK) measured enzyme activity in supernatants from treated and untreated hPDLCs following the manufacturer’s protocol. pNPP served as the substrate, with its dephosphorylation detected via a colorimetric reaction. 90 µL of supernatants were pipetted into a 96-well plate and mixed with 50 µL of 5 mM pNPP solution. After 1 h of incubation at room temperature in the dark, 20 µL of stop solution was added. Absorbance was measured at 405 nm using a microplate reader.

### 4.10. Statistical Evaluation

All experiments were performed with at least three different donors. The data presented were evaluated as mean ± standard deviation (SD). Statistical significance was calculated with the one-way multifactorial variance analysis ANOVA test or *t*-test using graphed pad prism 9 (GraphPad Software Inc., San Diego, CA, USA). Statistical significance was assessed when * *p* < 0.05, ** *p* < 0.01, *** *p* < 0.001 and **** *p* < 0.0001 and documented in the figures.

## 5. Conclusions

This study successfully isolated and characterized human periodontal ligament cells (PDLCs), confirming their stem cell-like properties and regenerative capacity. Platelet-rich fibrin (PRF) exerted a dual effect by reducing PDLC proliferation while enhancing osteogenic differentiation, with stronger effects observed under direct treatment with PRF-conditioned medium compared to indirect application. These results indicate that PRF functions not only as a scaffold but also as a biologically active reservoir of growth factors capable of modulating PDLC behavior and creating a microenvironment favorable for periodontal repair. Clinically, these findings might support the use of PRF as a practical autologous biomaterial for regenerative dentistry. By promoting osteogenic differentiation while supporting wound healing, PRF may enhance the predictability and effectiveness of periodontal regenerative procedures.

## Figures and Tables

**Figure 1 ijms-26-09459-f001:**
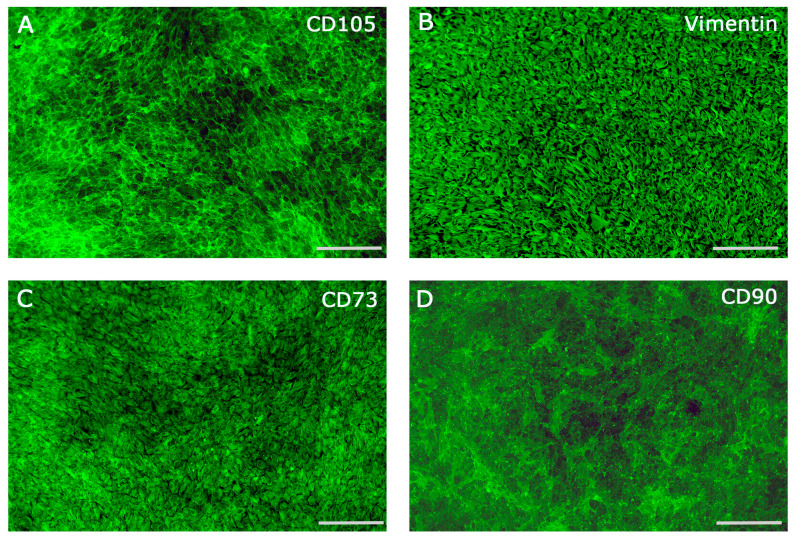
Immunofluorescence staining of hPDLCs for key mesenchymal and periodontal ligament-associated markers CD105 (**A**), Vimentin (**B**), CD73 (**C**) and CD90 (**D**). Scale bars 250 µm.

**Figure 2 ijms-26-09459-f002:**
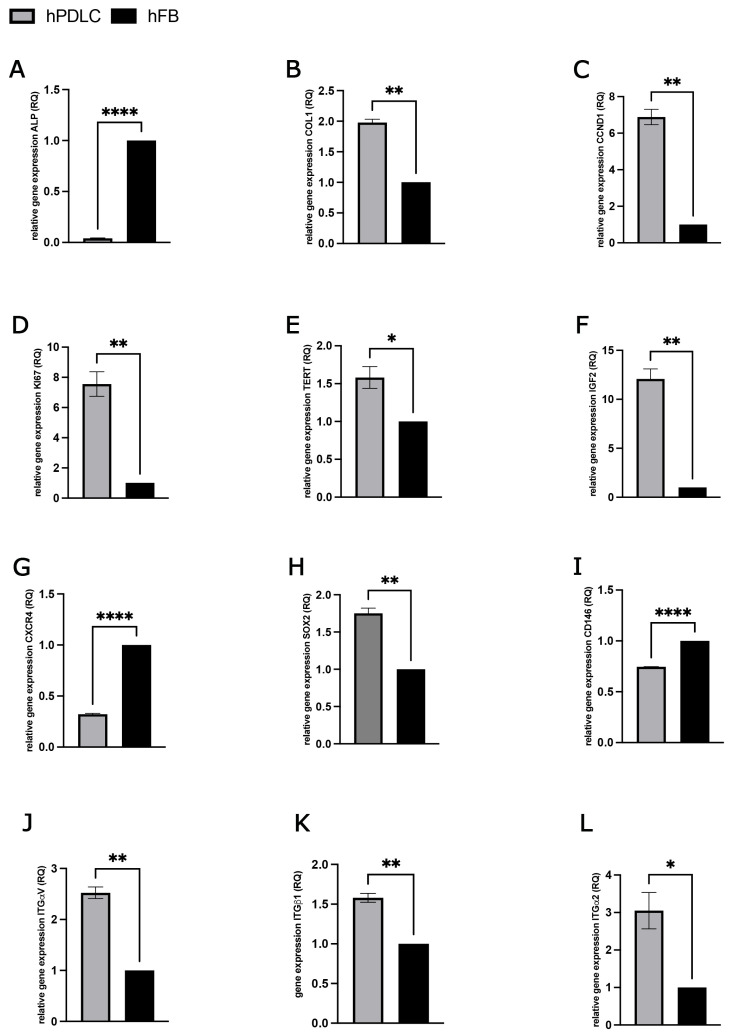
Gene expression analyses of markers associated with osteogenesis, cell proliferation, cell cycle regulation, stemness, and cell adhesion in hPDLCs (gray columns) compared to hFBs (black columns). (**A**) ALP, (**B**) COL1, (**C**) CCND1, (**D**) KI67, (**E**) TERT, (**F**) IGF2, (**G**) CXCR4, (**H**) SOX2, (**I**) CD146, (**J**) ITGα2, (**K**) ITGβ1, and (**L**) ITG αV. Statistical significance was assessed when * *p* < 0.05, ** *p* < 0.01 and **** *p* < 0.0001.

**Figure 3 ijms-26-09459-f003:**
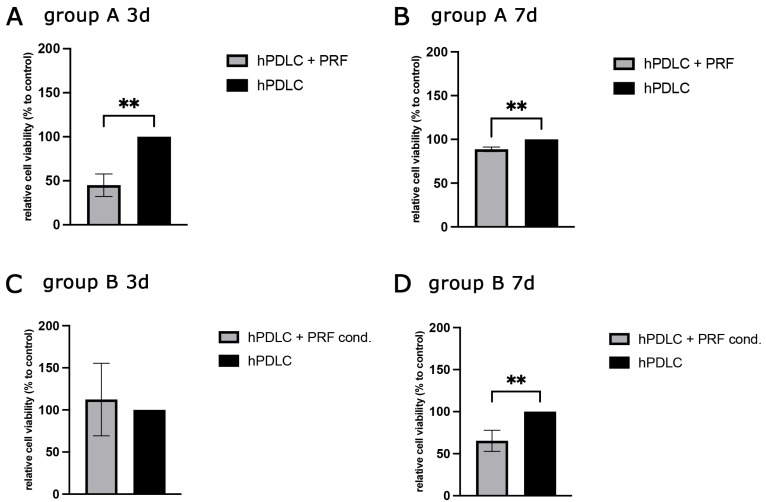
PRF-mediated effect on hPDLCs cell viability comparing two different PRF applications and different time points of treatment. (**A**,**B**) group A 3d and 7d (**C**,**D**) group B 3d and 7d of treatment. Statistical significance was assessed when ** *p* < 0.01.

**Figure 4 ijms-26-09459-f004:**
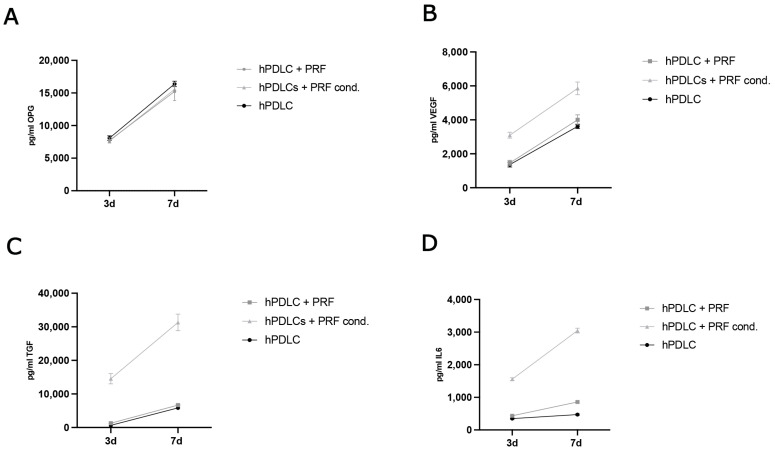
Evaluation of protein concentration in cell culture supernatants in hPDLCs and PRF treated hPDLCs after 3d and 7d of treatment. (**A**) OPG, (**B**) VEGF, (**C**) IL6, and (**D**) TGF.

**Figure 5 ijms-26-09459-f005:**
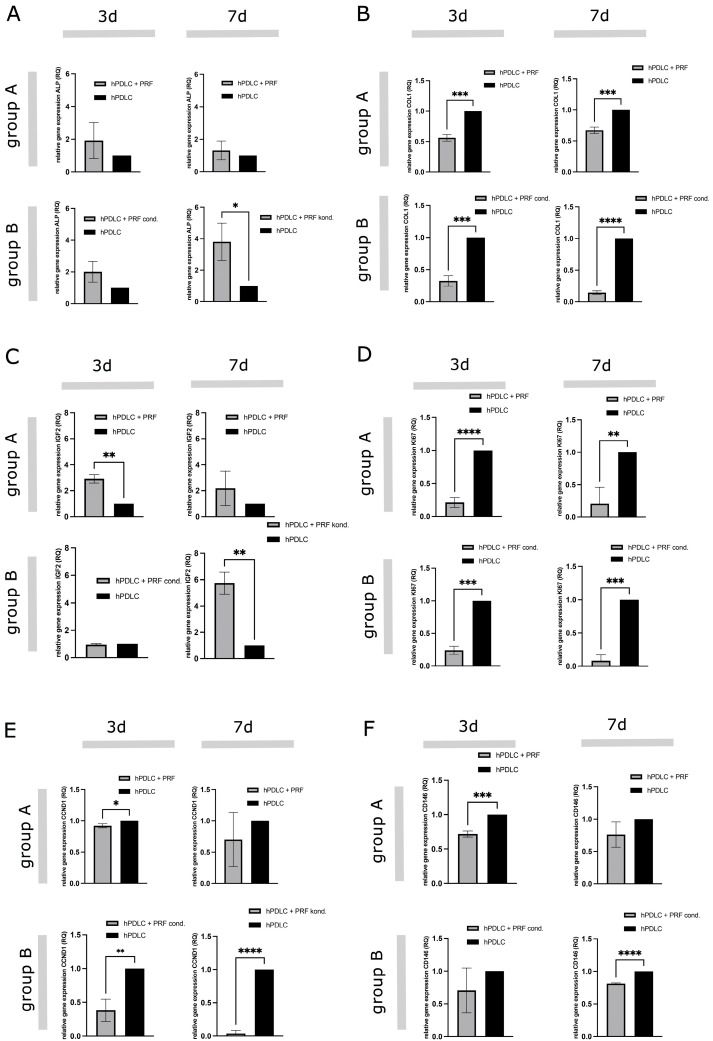
PRF-mediated effect on gene expression of differentiation, proliferation and stem cell associated markers in hPDLCs. Gene expression analyses of markers associated with osteogenesis, cell proliferation, cell cycle regulation, stemness, and cell adhesion in hPDLCs compared to hFBs. (**A**) ALP, (**B**) COL1, (**C**) IGF2, (**D**) KI67, (**E**) CCND1, (**F**) CD146. Statistical significance was assessed when * *p* < 0.05, ** *p* < 0.01, *** *p* < 0.001 and **** *p* < 0.0001.

**Figure 6 ijms-26-09459-f006:**
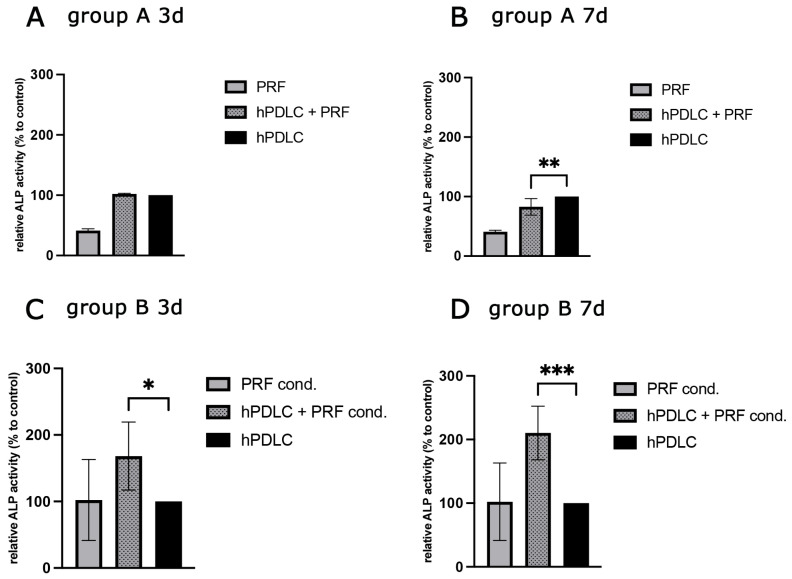
Alkaline phosphatase activity in PRF treated hPDLCs comparing two different PRF applications and different time points of treatment. (**A**,**B**) group A 3d and 7d (**C**,**D**) group B 3d and 7d of treatment. Statistical significance was assessed when * *p* < 0.05, ** *p* < 0.01 and *** *p* < 0.001.

## Data Availability

The data that support the findings of this study are available on request from the corresponding author (E.D.).

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
