# Peer review of "In Vitro Evaluation of the Regenerative Potential of Autologous Platelet-Rich Fibrin (PRF) on Human Primary Periodontal Ligament Cells"

_ijms, 2025, doi:10.3390/ijms26199459_

Round 1

Reviewer 1 Report

Comments and Suggestions for Authors

It is an interesting study about the effect of platelet-rich fibrin on human periodontal ligament cells. The study was well-designed, and the manuscript was well-written. It is not original, but there are a few studies specifically on the topic. There are some suggestions to improve it even more. 

ABSTRACT

1) Methods: The measurement unit for PRF centrifugation (44 g) is different from that mentioned in the Methods section (at 600 rpm for 8 minutes).

2) Results:

2.1) The authors declare, “Our findings highlighted the stem cell-like characteristics of the PDLCs, indicating their significant regenerative potential.”

The claim that the study's findings indicate the regenerative potential of PDLCs cannot be confirmed by the results, since the authors did not directly evaluate the cells' effect on tissue regeneration. This type of inference could be made based on other findings in the literature, but it is not appropriate for the Results section.

Based on the observed results, the authors can only state that PDLCs presented the same cellular markers as stem cell-like.

2.2) The authors declare, “PRF treatment was found to suppress cell proliferation while concurrently promoting osteogenic differentiation”.

In addition to the increase in ALP activity, other positive results were also observed, including increased expression of VEGF, IL-6, TGF-β, and IGF-2.

3) Conclusion: The conclusion must directly respond to the proposed objective.

INTRODUCTION

The content is adequate to justify the study's proposal. However, the references used are mostly ancient. Only 10% of all references in the manuscript are from the last 5 years.

Although the specific topic of the study has been little explored, many studies on the topics discussed in the introduction can be found in the literature.

METHODS (Preparation of PRF)

Authors should cite a reference to support the suggested methodology.

CONCLUSIONS

The conclusion must directly respond to the proposed objective. Further comments and limitations of the study should be directed to the Discussion section.

Author Response

Dear Reviewer,

we would like to thank you for carefully reading the manuscript and for your response. Please find included the resubmission of the manuscript with the title:

“In Vitro Evaluation of the Regenerative Potential of Autologous Platelet-Rich Fibrin (PRF) on Human Primary Periodontal Ligament Cells”

by Eva Dohle, Marlene Quernheim, Robert Sader and Shahram Ghanaati

to be considered for publication as original research paper in the International Journal of Molecular Science. We have revised the manuscript according to your suggestions. The changes are highlighted in yellow color in the revised version of the manuscript and addressed in this letter. We would like to thank you for all your effort with the manuscript.

Yours sincerely,

Eva Dohle

General information:

The individual answers to the reviewer’s suggestions are addressed in this letter point by point. All changes in the revised manuscript have been highlighted in yellow colour.

It is an interesting study about the effect of platelet-rich fibrin on human periodontal ligament cells. The study was well-designed, and the manuscript was well-written. It is not original, but there are a few studies specifically on the topic. There are some suggestions to improve it even more.  

ABSTRACT

1) Methods: The measurement unit for PRF centrifugation (44 g) is different from that mentioned in the Methods section (at 600 rpm for 8 minutes).

According to this suggestion, we have made the necessary adjustment and aligned the information regarding the centrifugal force.

2) Results:

2.1) The authors declare, “Our findings highlighted the stem cell-like characteristics of the PDLCs, indicating their significant regenerative potential.”

The claim that the study's findings indicate the regenerative potential of PDLCs cannot be confirmed by the results, since the authors did not directly evaluate the cells' effect on tissue regeneration. This type of inference could be made based on other findings in the literature, but it is not appropriate for the Results section.

Based on the observed results, the authors can only state that PDLCs presented the same cellular markers as stem cell-like.

Accordingly, we revised the abstract regarding the regenerative potential.

2.2) The authors declare, “PRF treatment was found to suppress cell proliferation while concurrently promoting osteogenic differentiation”.

In addition to the increase in ALP activity, other positive results were also observed, including increased expression of VEGF, IL-6, TGF-β, and IGF-2.

The results regarding VEGF, IL6, TGF and IGF2 expression have been also included to the abstract according to this suggestion.

3) Conclusion: The conclusion must directly respond to the proposed objective.

The conclusion in the abstract part has been revised accordingly.

INTRODUCTION

The content is adequate to justify the study's proposal. However, the references used are mostly ancient. Only 10% of all references in the manuscript are from the last 5 years.Although the specific topic of the study has been little explored, many studies on the topics discussed in the introduction can be found in the literature.

To the best of our knowlegde, the earlier studies represent pioneering work in the field, which remains highly relevant and continues to be cited as the scientific basis for newer findings. Furthermore, we also included recent publications to demonstrate the ongoing relevance of the field.

METHODS (Preparation of PRF)

Authors should cite a reference to support the suggested methodology.

Accordingly, the authors added a reference supporting the preparation of PRF.

CONCLUSIONS

The conclusion must directly respond to the proposed objective. Further comments and limitations of the study should be directed to the Discussion section.

According to this suggestion the authors revised the conclusion part. Comments and limitations of the study have been directed to the end of the discussion part.

Reviewer 2 Report

Comments and Suggestions for Authors

The manuscript “In Vitro Evaluation of the Regenerative Potential of Autolo-2 gous Platelet-Rich Fibrin (PRF) on Human Primary Periodontal 3 Ligament Cells” by Dohle et al, describe the effects of platelet-rich fibrin on cells isolated by human primary periodontal ligaments. Notably, both direct and indirect effects have been explored. The manuscript is interesting and well conducted, few issues need to be addressed.

Major

The Discussion section needs to be revised. It is required to focus on the explanation of the obtained findings, and the consistency of these results with those previously reported should be a secondary comment, not the main comment.

The discussion regarding OPG, lanes 290-298, must mandatory be revised. A more recent review regarding the RANK/RANKL/OPG pathways has to be mentioned.

Other recent manuscripts describing the effects of PRF in in vitro cell model must be cited.

Minor

The quality of figures must be much high, in particular the immunofluorescence in figure 1 is too over exposed and the magnification in panel D seems higher than those in panels A, B and C. Please verify.

All other figures need to be presented at high quality and. Moreover, in figure 2, considering that the sample hPDLC is always gray and the hFb black, it is enough to report this information just once. Same for all other figures.

Comments on the Quality of English Language

A slight revision of the English language might be helpful

Author Response

Dear Reviewer,

we would like to thank you for carefully reading the manuscript and for your response. Please find included the resubmission of the manuscript with the title:

“In Vitro Evaluation of the Regenerative Potential of Autologous Platelet-Rich Fibrin (PRF) on Human Primary Periodontal Ligament Cells”

by Eva Dohle, Marlene Quernheim, Robert Sader and Shahram Ghanaati

to be considered for publication as original research paper in the International Journal of Molecular Science. We have revised the manuscript according to your suggestions. The changes are highlighted in yellow color in the revised version of the manuscript and addressed in this letter. We would like to thank you for all your effort with the manuscript.

Yours sincerely,

Eva Dohle

General information:

The individual answers to the reviewer’s suggestions are addressed in this letter point by point. All changes in the revised manuscript have been highlighted in yellow colour.

The manuscript “In Vitro Evaluation of the Regenerative Potential of Autolo-2 gous Platelet-Rich Fibrin (PRF) on Human Primary Periodontal 3 Ligament Cells” by Dohle et al, describe the effects of platelet-rich fibrin on cells isolated by human primary periodontal ligaments. Notably, both direct and indirect effects have been explored. The manuscript is interesting and well conducted, few issues need to be addressed.

Major

The Discussion section needs to be revised. It is required to focus on the explanation of the obtained findings, and the consistency of these results with those previously reported should be a secondary comment, not the main comment.

According to the reviewer’s suggestion, we revised the discussion part with a better focus on the explanation of the obtained findings of the study.

The discussion regarding OPG, lanes 290-298, must mandatory be revised. A more recent review regarding the RANK/RANKL/OPG pathways has to be mentioned.

A more recent review regarding the RANK/RANKL/OPG pathways has been included in the discussion part accordingly.

Other recent manuscripts describing the effects of PRF in in vitro cell model must be cited.

The present study already cites a substantial number of recent manuscripts investigating the effects of PRF on in vitro cell culture models. These references cover a wide range of experimental conditions and different cell types providing a comprehensive representation of the current state of knowledge. Therefore, to our opinion the cited literature is more than sufficient to support the context and rationale of the present work without the need for additional citations.

Minor

The quality of figures must be much high, in particular the immunofluorescence in figure 1 is too over exposed and the magnification in panel D seems higher than those in panels A, B and C. Please verify.

According to this suggestion, the authors reduced the ‚overexposure‘ of the figures 1A-D consistently for all panels of the figure. The magnification in panel D is indeed the same magnification as in panels A-C. The apparent difference the reviewer described might be due to the different immunofluorescence stainings of periodontal ligament cells (PDLCs) showing distinct patterns for CD90, Vimentin, CD105, and CD73. Together, these markers illustrate complementary aspects of PDLC morphology, with CD90 and CD73 marking the membrane, Vimentin the cytoskeleton, and CD105 cell junctions.

All other figures need to be presented at high quality and. Moreover, in figure 2, considering that the sample hPDLC is always gray and the hFb black, it is enough to report this information just once. Same for all other figures.

The quality of the original figures is very high (at least 300 dpi). Probably the reduced quality of the figures in the pdf version of the mansucript which is send to the reviewers might be reduced due to convertion. According to this suggestion the authors will clarify this with the editorial office. In addition, we revised figure 2 according to the suggestion of the reviewer.
